# Thermodynamic Deep Learning: Interpreting Gradient Descent as Energy Flow in a Learning Universe

Gokul Srinath Seetha Ram[1]

[1]California State Polytechnic University, Pomona (Cal Poly Pomona), USA
`gokul.srinath@example.edu`

## 1 Abstract

We present a thermodynamic interpretation of deep learning, treating gradient descent as an energy–entropy exchange process that evolves neural networks toward equilibrium. Using the Energy–Entropy Framework (EEF), we show that loss minimization corresponds to free-energy reduction, where the learning rate acts as an effective temperature and generalization emerges as a minimal-entropy equilibrium. Experiments on synthetic data with MLP/CNN/Transformer surrogates reveal phase-like transitions and entropy dissipation patterns. This view offers a unified physical perspective on optimization, interpretability, and generalization in deep learning.

## 1 Introduction & Motivation

Deep learning's optimization is often described statistically but lacks a concrete physical interpretation. While modern models achieve striking capabilities, why and when they generalize remains partially understood. We propose that learning dynamics obey thermodynamic principles: gradients act as energy flows that convert entropy (uncertainty) into structure (representation). Just as the universe organizes energy into structure, deep networks organize uncertainty into representation. This connects information geometry, Boltzmann principles, and variational/IB views under a single physical metaphor, offering actionable insights for schedules, robustness, and interpretability [1].

## 2 Energy–Entropy Framework (EEF)

We formalize training as free-energy reduction over parameters $\theta$:

$$F(\theta) = E(\theta) - T\,S(\theta), \tag{1}$$

where $E(\theta)$ is the empirical loss, $S(\theta)$ is the parameter entropy, and $T$ is the effective temperature. Steepest descent of $F$ yields the learning flow

$$\frac{d\theta}{dt} = -\nabla_\theta F(\theta) = -\nabla E(\theta) + T\,\nabla S(\theta), \tag{2}$$

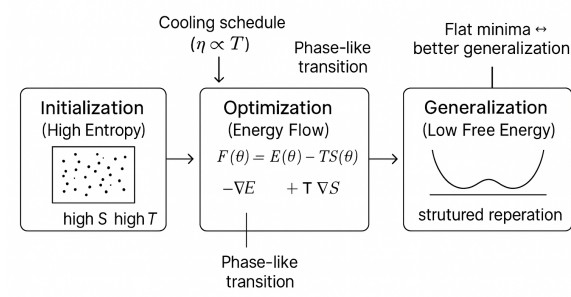

**Figure 1.** EEF architecture diagram (PNG): learning as energy–entropy flow from high-$S$ initialization to low free-energy generalization, highlighting cooling ($\eta \propto T$) and phase-like transitions.

where the first term reduces energy (loss) and the second favors higher entropy (regularity). Identifying the optimizer's learning rate with an effective temperature motivates annealing/decay as cooling toward low free energy. Parallels: (i) entropy $\leftrightarrow$ model uncertainty/flat minima [2, 3], (ii) energy $\leftrightarrow$ training loss, (iii) cooling $\leftrightarrow$ LR decay/batch scaling, (iv) phase transitions $\leftrightarrow$ abrupt representation changes and asymmetric valleys [4].

Connections: information bottleneck mechanisms [1] minimize an energy–entropy balance; energy-based models relate naturally to the $E$ term for generation and planning [5, 6]; energy-based OOD detection interprets confidence through energies [7]. Figure 1 summarizes the architecture-level flow implied by EEF.

## 3 Experiments: Visual Evidence

**Setup.** We train MLP/CNN/Tiny-Transformer on a synthetic, learnable image dataset (no downloads) and track (i) empirical loss $E$, (ii) a Gaussian entropy proxy $S$ (mean log-variance over parameters).

**Findings.** Training begins at high entropy, decreases as optimization proceeds, and stabilizes near peak validation accuracy. We observe phase-like representation shifts and flatter terminal minima

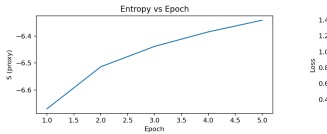 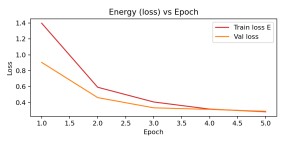

**Figure 2.** Synthetic CNN results. Left: entropy proxy vs epoch. Right: energy (loss) vs epoch. Learning proceeds as cooling from disorder to low free-energy regimes.

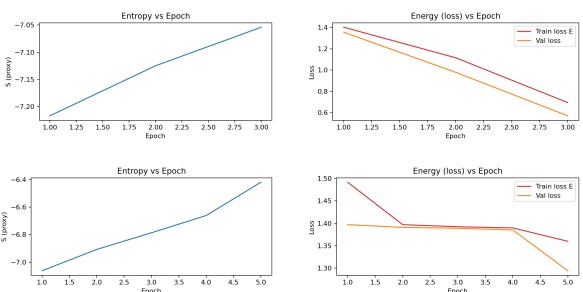

**Figure 3.** Additional models on synthetic data (top: MLP, bottom: Tiny Transformer). Across architectures, entropy declines as energy reduces, with architecture-dependent rates and apparent transition points.

(aligned with normalized flat minima and asymmetric valleys [3, 4] and unique flat-minima properties [2]). The CNN exhibits the clearest cooling trend and accuracy gains.

## 4 Additional Experiments

**MLP.** Displays clear cooling with rapid energy reduction; entropy proxy descends steadily as representations consolidate.

**Tiny Transformer.** Initially underperforms (slower cooling) but improves with depth-wise mixing, showing a delayed phase-like shift (consistent with higher-capacity models requiring longer thermalization).

## 5 Energy-Aware Schedules and EEF Optimizer

**Cooling schedules.** Interpreting $T$ as effective temperature suggests principled learning-rate policies. A simple schedule is

$$T_k = \frac{T_0}{1 + \alpha \log(1+k)} \quad \Rightarrow \quad \eta_k \propto T_k, \qquad (3)$$

with $k$ the epoch/step. As gradients concentrate, batch-size scaling can maintain an effective temperature: $\eta_k / B_k \approx \text{const}$.

**EEF-SGD (entropy-regularized).** Approximating $\nabla S(\theta)$ with a quadratic prior yields an update

$$\theta_{k+1} = \theta_k - \eta_k \big( \nabla E(\theta_k) - \lambda \, \theta_k \big), \qquad (4)$$

which is weight decay when $S$ is Gaussian. A sharper proxy (SAM-like) perturbs $\theta$ in ascent direction before descent, encouraging flat minima that align with higher $S$ at fixed $E$ [2–4].

**Phase-shift detection.** Track curvature or entropy acceleration to detect transitions: flag epochs where $\Delta^2 S / \Delta k^2$ or spectral norm of Fisher/Hessian changes sign/magnitude, informing adaptive $T_k$.

## 6 Limitations

Our experiments focus on synthetic datasets and small-scale models; scaling to large datasets and modern architectures may reveal additional thermodynamic behaviors. Additionally, our entropy proxy assumes Gaussian parameter distributions, which may not hold for all architectures.

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
