# OpenReview forum: "Thermodynamic Deep Learning: Interpreting Gradient Descent as Energy Flow in a Learning Universe"
_NLDL.org/2026/Abstracts_Track — NLDL 2026 Abstracts_

### Official Review · Reviewer_zSNs · 2025-10-31

**Soundness:** 2
**Correctness:** 2
**Rating:** 4
**Confidence:** 2

**Summary:**

The authors introduce a interpretation of gradient decent based deep learning through the lens of thermodynamics. At first glance the idea is interesting and timely, however the evidence to support based on synthetic data it is very weak.

**Strengths:**

- It is an interesting perspective to reconcile deep learning with physics inspired learning.
- Some analogues make intuitive sense (e.g. energy as loss, entropy as parameter entropy, and learning rate as temperature)

**Weaknesses:**

- “Parameter entropy” computed as mean log-variance of weights is a questionable proxy for entropy
-Experiments are only on synthetic data with small models. The claims of “phase transitions” and “entropy dissipation” are qualitative and rely on this questionable entropy proxy.
- There are no details about the synthetic data.

---

### Official Review · Reviewer_xLGN · 2025-11-01

**Soundness:** 2
**Correctness:** 2
**Rating:** 2
**Confidence:** 4

**Summary:**

The paper proposes a thermodynamic interpretation of deep learning where gradient descent is viewed as an energy-entropy exchange process. The authors formalize training as free-energy minimization, where loss acts as energy, parameter entropy as disorder, and learning rate as effective temperature. Experiments on synthetic datasets with MLP/CNN/Transformer architectures show entropy declining during training, which the authors interpret as "cooling" toward equilibrium states.

**Strengths:**

**TL;DR**: The paper provides an interesting physical metaphor connecting optimization dynamics to thermodynamic principles and presents a unified view linking information geometry, flat minima research, and energy-based models.

**Long form**:
- (major) Provides a cohesive framework (EEF) that connects several existing concepts (information bottleneck, flat minima, energy-based models) under a single thermodynamic interpretation, potentially offering intuitive insights for practitioners
- (minor) Clear presentation with helpful visualizations showing entropy and energy trajectories across different architectures
- (minor) Proposes concrete, actionable schedules (Eq. 3) and phase-shift detection mechanisms based on the thermodynamic view

**Weaknesses:**

**TL;DR**: The contributions are very unclear as the main components (entropy regularization, flat minima preference, thermodynamically-inspired schedules) are well-established. Critical experimental validation is compromised by using an undisclosed synthetic dataset.

**Long form**:
- (major) Lack of novelty: The paper reframes existing techniques without clear added value. Weight decay as Gaussian entropy regularization (Eq. 4) is standard; flat minima improving generalization is well-known [2-4]; SAM already encourages high-entropy flat regions.
- (major) No new algorithmic contribution: The proposed EEF-SGD (Eq. 4) reduces to standard weight decay. The "thermodynamic" interpretation doesn't lead to new optimization methods that outperform existing approaches
- (major) Prior work on thermodynamic learning: Learning rate schedules inspired by physics already exist (e.g., SGDR/cosine annealing from simulated annealing, cyclical learning rates). The paper doesn't clearly distinguish its contribution from these
- (major) Non-reproducible experiments: Uses an undisclosed "synthetic, learnable image dataset (no downloads)" making results impossible to verify or reproduce. This violates basic scientific standards
- (major) Weak empirical validation: Only toy synthetic experiments on small models. No comparison with baseline schedules, no ablations, no large-scale validation to demonstrate whether the thermodynamic view provides practical benefits
- (major) Questionable entropy proxy: Assumes Gaussian parameter distributions for entropy calculation (acknowledged in §6), which may not hold, making the "thermodynamic" measurements potentially meaningless
- (minor) The metaphor linking "universe organizing energy" to "networks organizing uncertainty" (§1) is vague and doesn't add scientific rigor
- (minor) Missing comparisons: How does the proposed T_k schedule (Eq. 3) compare quantitatively to standard cosine/step decay schedules?
- (minor) Phase transitions are mentioned but not rigorously defined or detected algorithmically
- (minor) The effective temperature interpretation of learning rate, while intuitive, isn't validated (e.g., does it predict generalization better than existing theory?)
- (minor) No discussion of when/why the thermodynamic analogy might break down or be misleading

---

### Official Review · Reviewer_v3uF · 2025-11-04

**Soundness:** 3
**Correctness:** 3
**Rating:** 4
**Confidence:** 3

**Summary:**

The work discusses an energy-entropy framework, which poses the optimization task (e.g. as used in deep learning) as an energy flow and entropy reduction. Experiments based on the framework demonstrate the empirical relationship of variation in the loss and variation in the entropy of the parameters.

**Strengths:**

The thought process behind the work is interesting an insightful. The introduction mentions "gradients act as energy flows that convert entropy (uncertainty) into structure (representation)" and "deep networks organize uncertainty into representation". Thus, entropy of the

The paper draws parallels between thermodynamic concepts and some of the model/algorithmic aspects, which is the basis of the proposed framework and analysis.

The experiments are limited but targeted, and show the energy-entropy relationship for 3 different DL models.

**Weaknesses:**

In the primary equation (1), while E and S have a clear interpretation, the interpretation or parallel of F in terms of an algorithmic counterpart is not discussed. Thus, the role of F in terms of the energy minimization of a loss function is not clear.
In optimization, one typically operates with only E. Hence, bringing in the notion of S is indeed interesting, but unless the role of F in the parametric system is clear, the overall interpretation of the optimization in terms of thermodynamic is still limited.

There have been other models relating thermodynamic principles to ML / optimization frameworks e.g. annealing, diffusion, frameworks based on Gibbs energy etc. Perhaps, the author can briefly discuss how this work stands out over other such frameworks.

Some insight on the differences of the entropy minimization trends across different DL frameworks may lead to some new directions of thought in terms of assessing different models.

---

### Decision · Program_Chairs · 2025-11-05

**Decision:**

Accept

**Comment:**

The reviewers found the abstract borderline, yet the PCs believe it will be of interest to the community and should have the opportunity be presented.